# Effects of *Lactobacillus plantarum* and *Weissella viridescens* on the Gut Microbiota and Serum Metabolites of Mice with Antibiotic-Associated Diarrhea

**DOI:** 10.3390/nu15214603

**Published:** 2023-10-30

**Authors:** Zhiwei Yan, Zhuangzhuang Liu, Yong Ma, Zhao Yang, Gang Liu, Jun Fang

**Affiliations:** 1College of Bioscience and Biotechnology, Hunan Agricultural University, Changsha 410128, China; yan-zhiwei@stu.hunau.edu.cn (Z.Y.); 609543294@stu.hunau.edu.cn (Z.L.); mayong@stu.hunau.edu.cn (Y.M.); yzj991223@stu.hunau.edu.cn (Z.Y.); gangle.liu@gmail.com (G.L.); 2Hunan Provincial Engineering Research Center of Applied Microbial Resources Development for Livestock and Poultry, College of Bioscience and Biotechnology, Hunan Agricultural University, Changsha 410128, China

**Keywords:** *Lactobacillus plantarum*, *Weissella viridescens*, antibiotic-associated diarrhea, microbiota, metabolism

## Abstract

Antibiotic-associated diarrhea (AAD) refers to diarrhea caused by gut microbiota disorders after the use of antibiotics, which seriously threatens the health of humans and animals. Therefore, it is necessary to find an effective therapy to treat AAD. This research aimed to explore the effects of *Lactobacillus plantarum* H-6 (*L. plantarum* H-6) and *Weissella viridescens* J-1 (*W. viridescens J-1*) on alleviating antibiotic-associated diarrhea induced by lincomycin hydrochloride (LH) in mice. The results show that *L. plantarum* H-6 could significantly reduce the expression of pro-inflammatory factors such as *IL-1β* and *IL-6* in colon tissue. At the same time, *L. plantarum* H-6 significantly increased the abundance of *Lactobacillus* and *Akkermansia*, decreased the abundance of *Bacteroides*, and increased the contents of L-tryptophan, LysoPC (20:4 (8Z, 11Z, 14Z, 17Z)), reduced riboflavin, threoninyl–methionine, and N-palmitoyl in serum. However, *W. viridescens* J-1 had little effect on the treatment of AAD. It can be concluded that *L. plantarum* H-6 can regulate mice’s colonic microbial composition, improve their serum metabolic process, and alleviate antibiotic-associated diarrhea. This research may provide a novel therapeutic option for AAD.

## 1. Introduction

AAD is caused by a gut microbiota imbalance and disorders after antibiotic treatment. It is one of the most frequent side effects of antibiotic therapy [1,2,3]. AAD is a complex disease related to gut microbiota, infection, and other clinical factors (especially antibiotic treatment). It is characterized by the destruction of gut microbiota, a decrease in intestinal short-chain fatty acid (SCFA) concentration, the accumulation of carbohydrates and bile acid in the colon, and a change in water absorption [4,5]. In recent years, the incidence of AAD in patients treated with antibiotics has slowly increased; in some cases, it has even reached 30% [6,7] and has shown an increasing trend with the upgrading and replacement of antibiotics. More than 20% of AAD patients have experienced ineffectiveness of the first treatment, and 40–60% have recurrent symptoms of diarrhea [8]. The incidence of AAD is higher in patients that are admitted to intensive care units, where there is a possibility of increasing other infections, which can lead to higher medical costs and mortality [8,9]. Therefore, the prevention and treatment of AAD is particularly important.

Although the pathogenesis of AAD has not been fully elucidated, it is currently believed that antibiotics destroy gut microbiota, leading to secondary pathogen colonization and proliferation [10,11]. Therefore, supplementation and regulation of gut microbiota is one of the primary means of preventing and treating AAD. The prominent function of probiotics is to prevent and alleviate gastrointestinal diseases [12]. There is mass evidence that probiotics have a significant positive effect on maintaining the stability of intestinal flora. Dong Yang et al. showed that lactic acid bacteria can be used as potential probiotics to strengthen the balance of gastrointestinal flora [13]. Oral *Bifidobacterium* can effectively improve the balance of intestinal flora in children with recurrent respiratory tract infection and reduce the frequency of recurrence [14]. In addition, probiotics play an important role in inhibiting the colonization of pathogenic bacteria in the intestine, which they achieve mainly through competitive inhibition and the production of antibacterial substances [15,16]. For example, *Lactobacillus reuteri* JCM1081 and TM105 can compete with *Helicobacter pylori* for gangliotetraosylceramide and sulfatide binding sites in gastric epithelial cells, thereby inhibiting the early colonization of *Helicobacter pylori* [17]. Another study showed that probiotics can secrete antibacterial substances (organic acids and bacteriocins) that cause steric hindrance and competitive adhesion sites to prevent *Helicobacter pylori* from binding to intestinal epithelial cells [18]. It can be seen that AAD has a tremendous negative impact on the intestinal health of humans and animals, and probiotics have the ability to regulate the structure of intestinal microbiota and promote intestinal health. Therefore, this study investigated whether *L. plantarum* H-6 and *W. viridescens* J-1 can regulate the intestinal microbial community structure and reduce the risk of AAD-related diarrhea in mice.

At present, there are many reports that probiotics can prevent AAD. Qin Guo et al. and Jan Lukasik et al. showed that the probability of AAD in children administered probiotics was lower than that in the control group [3,19]. In a study of adults, probiotic administration reduced the incidence of AAD by 38% [20]. In addition, Liying Zhang et al. showed that the prevalence of AAD in the elderly was low when probiotics were administered within two days of antibiotic treatment [21]. It can be seen that probiotic administration has a certain effect on AAD, but can probiotic administration effectively treat AAD? There are few reports on this aspect. In this study, an AAD mouse model was established by intragastric administration of lincomycin hydrochloride to explore the effects of *L. plantarum* H-6 and *W. viridescens* J-1 on antibiotic-associated diarrhea. The study aims to provide a new theoretical foundation for the treatment of AAD. *L. plantarum* H-6 increased the variety and abundance of gut microbiota in mice, decreased the contents of *IL-6* and *IL-1β*, and increased the secretion of *IL-10* and *TGF-β*. The results of serum metabolomics in mice show that *L. plantarum* H-6 improved AAD by regulating tryptophan metabolism. However, the improvement of *W. viridescens* J-1 was not as significant as that of *L. plantarum* H-6.

## 2. Materials and Methods

### 2.1. Bacterial Strains and Culture

*L. plantarum* H-6 and *W. viridescens* J-1 were obtained from the Engineering Research Center of Applied Microbial Resources Development for Livestock and Poultry, Hunan Agricultural University, Changsha, China. The 16S rRNA sequences of the strains was uploaded to GenBank, and the sequence accession numbers of *L. plantarum* H-6 and *W. viridescens* J-1 were OQ692128 and OQ92138, respectively. The strains were activated by MRS broth medium, and 12 mL of activated strains were centrifuged at 8000 rpm (RCF = 6150 g) for 5 min at 4 °C. The bacteria were collected and resuspended in 5 mL of sterile saline.

### 2.2. Animals and Experimental Design

The Animal Ethics Committee of Hunan Agricultural University approved the animal experiment of this study (Approval number: NO. 202034; approval date: 15 April 2020). Female ICR mice were purchased from the Shanghai Slake Experimental Animal Co., Ltd. (Changsha, China). The mice were housed at a temperature of 25 ± 1 °C in a 12 h alternating lighting control room (maintaining a bright state from 8 a.m. to 8 p.m.) with full access to water and food.

After acclimatizing the animals, 24 mice were randomly distributed into four groups: a normal control group (NC), an antibiotic-associated diarrhea group (AAD), an *L. plantarum* H-6 treatment group (AAD + LP), and a *W. viridescens* J-1 treatment group (AAD + WV). NC group mice were given physiological sodium (0.2 mL) by gastric gavage, whereas the rest were all treated with lincomycin hydrochloride (LH) (300 mg/kg.BW) by gastric gavage at 8:30 a.m. for 7 days. Afterward, the mice of the AAD + LP and AAD + WV groups received 5 × 10^8^ CFU (0.2 mL) of *L. plantarum* H-6 and *W. viridescens* J-1 (in saline) via oral gavage for 7 days. Meanwhile, the mice of the AAD group were given physiological saline solution (0.2 mL) by gastric gavage for 7 days. The body weight, food consumption, and water intake of the mice were estimated every day. The diarrhea status of all mice was evaluated according to the scoring criteria (Table 1) of previous studies [22,23]. At the end of the treatment, all mice were fasted for 12 h, and blood was collected from the eyeballs of the mice, which were sacrificed by cervical dislocation. Colon segments were harvested and fixed in paraformaldehyde for 4% histopathological analysis.

### 2.3. DNA Isolation and Bioinformatics Analysis

The extraction of bacterial genomic DNA from colon contents was conducted according to the operating instructions of the Bacteria Genomic DNA Extraction Kit (Beijing Labgic Technology Co., Ltd., Beijing, China). The conserved region (V3–V4) of the bacterial 16S rRNA gene sequence was amplified by PCR using particular primers (Appendix A).

The PCR product was detected on the Illumina NovaSeq platform using the paired-end method. The initial data were merged using FLASH [24] (version 1.2.11, Baltimore, MD, USA) software. The quality was filtered using Trimmatic [25] (version 0.33, Jülich, Germany.) tools, and the chimeras were removed through UCHIME [26] (version 8.1, Birmingham, UK) to obtain high-quality tags sequences, which were used for further analysis. All of the processed sequences were clustered in OTUs with 97% similarity using USEARCH [27] (version 10.0, Tiburon, CA, USA), and α diversity was analyzed and determined using mothur [28] (version v.1.30, Ann Arbor, MI, USA). the QIIME program was used to create species abundance at various taxonomic levels, and the R language tool was utilized to construct the species composition map at different taxonomic levels. Subsequently, the previously obtained data were uploaded to online tools (http://huttenhower.sph.harvard.edu/lefse/) (accessed on 12 September 2022) for LefSe analysis, which was implemented to find biomarkers between different groups.

### 2.4. RT-qPCR

Total RNA was isolated from colon tissue with the TransZol Up RNA Kit (TransGen Biotech, Beijing, China) following the manufacturer’s instruction. Next, RNA was reverse transcribed into cDNA using the *Evo M-MLV* RT Mix Kit (Accurate Biology, Changsha, Hunan, China). Gene-specific primers (Appendix A) were generated by Sangon Biotech Co., Ltd. (Shanghai, China). The RT-qPCR experiment was conducted using the SYBR Green Premix Pro Taq HS qPCR Kit. The RT-qPCR was used to detect the content of *TGF-β* and interleukin (*IL-1β*, *IL-6*, *IL-10*) in colon tissue, and the *β-actin* gene was defined as the internal reference.

### 2.5. Pathological Analysis of Colon

This part of the experiment was completed according to the method in our previous study, with some alterations [29,30]. Briefly, 4% paraformaldehyde was used to fix the colon segments of the mice, which were then rinsed with normal saline. After dehydration using an ethanol gradient, all colon samples were embedded in paraffin. The normal experimental approach was then followed to stain 10 μm slices with hematoxylin and eosin (HE). Finally, the sections were observed using an optical microscope (Olympus BX41, Tokyo, Japan) under blind circumstances.

### 2.6. Serum Metabolomics

To assess the impact of the two bacteria on the serum metabolites of AAD mice, we extracted and analyzed the metabolites according to the experimental method by Suzhen Qi et al. [31]. The collected mouse blood was separated at 8000 rpm (RCF = 6150 g) for 10 min at 4 °C, and the recovered supernatant was placed in a 1.5 mL centrifuge tube. Next, 400 μL methanol/acetonitrile (1:1, *v*/*v*) pre-cooled to −80 °C were added to the centrifuge tube. After adequate shaking, the whole mixture was sonicated at 4 °C for 10 min before being left at 80 °C for 12 h. Subsequently, the entire system was centrifuged at 11,000 rotations per minute for 10 min at 4 °C, and the upper water phase was subjected to vacuum freeze-drying. Finally, the dried sample was redissolved in a methanol/acetonitrile mixed solvent. After ultrasonication, centrifugation, and freezing at −80 °C overnight, the sample was transferred to a brown bottle for further metabolomics analysis. Comparative metabolomics analysis of mouse serum in each treatment group was performed using the Shimadzu GC2030-QP2020 NX gas chromatograph–mass spectrometer (Shimadzu, Kyoto, Japan) equipped with an Agilent DB-5MS capillary column (30 m × 250 μm × 0.25 μm, J&W Scientific, Folsom, CA, USA). Refer to Qiangqiang Li et al. [32] for the detailed method.

### 2.7. Data Analysis

Spearman correlation analysis was performed on each mouse’s colonic microbial abundance and serum metabolite content using SPSS 22.0 (San Francisco, CA, USA). The analysis parameters were correlation bivariate. All data were analyzed using SPSS Statistics (version 22, San Francisco, CA, USA) and expressed as mean ± standard errors of the means (SEM). One-way analysis of variance (ANOVA) was used to compare differences between groups. Tukey’s multiple comparisons test was used to determine whether the differences were statistically significant. A *p*-value of less than 0.05 was considered statistically significant.

## 3. Results

### 3.1. Physiological Effects of Intragastric Administration of Lincomycin Hydrochloride

The diarrhea status score, weight change, and water intake of the mice are shown in Figure 1. The mice in the three lincomycin hydrochloride treatment groups exhibited weight loss, increased water intake, and different degrees of diarrhea, indicating that the AAD mouse model was successfully established. During the treatment period, compared with the AAD group, the diarrhea status score and water intake of the AAD + LP group decreased, whereas body weight increased. The mice in the AAD + WV group also showed similar phenomena, but the therapeutic effect was not as significant as in the AAD + LP group.

### 3.2. L. plantarum H-6 and W. viridescens J-1 Attenuate Intestinal Injury Induced by Lincomycin Hydrochloride

Histopathological section analysis of mouse colon tissue was performed after HE staining was completed. Histological changes in the different treatment groups are shown in Figure 2. The pathological appearance of colon tissue in the NC group was normal, and there was no infiltration of inflammatory cells, as predicted. In the AAD group, pathological analysis revealed a destroyed crypt structure and substantial inflammatory cell infiltration. After a week of *L. plantarum* H-6 treatment, colonic damage was restored and inflammatory infiltration was significantly reduced. Compared to the AAD group, the colonic crypts of the AAD + WV group reverted to a status similar to that of the NC group, and the inflammatory infiltration was basically restored; however, the effect was not as significant as that of the AAD + LP group.

### 3.3. Effects of L. plantarum H-6 and W. viridescens J-1 on Inflammatory Factors in Colon Tissue

AAD is usually accompanied by intestinal inflammation, which manifests as abnormal levels of inflammatory factors. The expression levels of cytokines (Figure 3) in the colon tissue of all groups were measured to determine whether *L. plantarum* H-6 and *W. viridescens* J-1 had anti-inflammatory effects. In contrast to those in the NC group, *IL-6* and *IL-1β* in the AAD group were considerably elevated (*p* < 0.05), whereas *IL-10* and *TGF-β* were dramatically decreased (*p* < 0.05). This indicates that excessive intake of antibiotics can stimulate the release of pro-inflammatory factors and result in inflammation. *IL-10* and *TGF-β* were elevated in the AAD + LP and AAD + WV groups, but *IL-6* and *IL-1β* were substantially lowered in the AAD + LP group (*p* < 0.05).

### 3.4. Effects of L. plantarum H-6 and W. viridescens J-1 on Serum Metabolomics in Mice

We conducted non-targeted metabolomics sequencing of mouse serum using HPLC/Q-TOF-MS to assess the impact of two probiotics on serum metabolites in mice. Utilizing the KEGG database, the common and different metabolites between groups were identified (Figure 4). Compared with the NC group, the AAD group up-regulated 405 differential metabolites and down-regulated 3062; compared with the AAD group, the AAD + LP group up-regulated 1690 differential metabolites and down-regulated 517; and the AAD + WV group up-regulated 1486 differential metabolites and down-regulated 719. In addition, a pathway analysis of the differential metabolites was conducted using MetaboAnalyst 5.0 (https://www.metaboanalyst.ca/MetaboAnalyst/) (accessed on 20 September 2022). The results show that tryptophan metabolism and glycerophospholipid metabolism were the most sensitive pathways in AAD mice after intragastric administration of *L. plantarum* H-6 and *W. viridescens* J-1 (Figure 4D,E). In addition, compared with the AAD group, riboflavin metabolism and glycosaminoglycan biosynthesis–chondroitin sulfate/dermatan sulfate metabolism in the AAD + LP and AAD + WV groups also changed significantly (*p* < 0.05).

Based on the above data, several metabolites were found to be significantly altered in AAD mice and reversed after probiotic supplementation. L-tryptophan, LysoPC (20:4 (8Z, 11Z, 14Z, 17Z)), reduced riboflavin, threoninyl–methionine, and N-palmitoyl serine were significantly decreased in AAD mice, whereas diethyl succinate was significantly increased (*p* < 0.05).

### 3.5. The Influence of L. plantarum H-6 and W. viridescens J-1 on the Composition and Diversity of Gut Flora

Intestinal microbial dysbiosis is one of the characteristics of AAD. We used the Illumina NovaSeq platform to study the effects of *L. plantarum* H-6 and *W. viridescens* J-1 on intestinal microbes in AAD mice. We used the Chao 1, ACE, and Shannon indexes to describe the richness and diversity of mouse colon microorganisms. The results of colonic microbial diversity in mice are shown in Figure 5. The above three indicators of the mice decreased significantly (*p* < 0.05) after intragastric administration of antibiotics. This demonstrates that antibiotic therapy might seriously damage the diversity of colon microorganisms in mice. In comparison to the AAD group, the three indexes of the AAD + LP group increased dramatically (*p* < 0.05), especially the Shannon index, which was close to the normal level. Unfortunately, we did not observe a similar result in the AAD + WV group. These results indicate that *L. plantarum* H-6 could have alleviated the loss of microbial diversity caused by lincomycin hydrochloride.

We further analyzed the microbial classification and relative abundance in mouse colons, and we determined the differences in the species composition of each group at the phylum and genus levels (Figure 6 and Appendix A). Firmicutes, Bacteroidetes, Proteobacteria, and Desulfobacterota were the major taxa in the NC group, and their relative abundances were 65.9%, 21.6%, 3.1%, and 5.6%, respectively. Significantly less Firmicutes and Desulfobacterota were present in the AAD group than in the NC group, but Bacteroidetes and Verrucomicrobiota increased considerably. The findings demonstrate that lincomycin hydrochloride might change the composition of intestinal flora in mice, which could be considered an intestinal microecology disorder. In the AAD + LP group, the relative abundances of Firmicutes, Bacteroidetes, Proteobacteria, and Desulfobacterota were 59.6%, 29.5%, 3.0%, and 1.3%, respectively. The disorders of Firmicutes, Bacteroidetes, and Desulfobacterota were reversed after *L. plantarum* H-6 treatment, but Proteobacteria was significantly lower than in the NC and AAD groups. Bacteroidetes levels were significantly higher in the AAD + WV group than in the NC and AAD + LP groups.

The top 10 bacteria at the genus level are shown in Figure 7. *Unclassified_Muribaculaceae*, *Lactobacillus*, *Bacteroides*, and *Lachnospiraceae_NK4A136_group* were the most numerous bacteria. The proportion of *unclassified_Muribaculaceae* was 9.5%, 10.6%, 15.4%, and 15.7%; that of *Lactobacillus* was 12.65%, 2.2%, 21.1%, and 13.5%; that of *Bacteroides* was 7.3%, 14.0%, 8.0%, and 12.8%; and that of *Lachnospiraceae_NK4A136_group* was 12.3%, 7.4%, 11.2%, and 8.8% in the NC, AAD, AAD + LP, and AAD + WV groups, respectively.

### 3.6. Correlation Analysis on Metabolites, Bacterium, and Immune Factors

Combining the results of serum metabolomics, microbial sequencing, and the content of cytokines in the colon tissue, we found a correlation between differential metabolites, differential intestinal strains, and cytokines. Figure 8 summarizes the complex relationship between the changes in key variables. In the correlation analysis of intestinal bacteria and factors, *Lactobacillus* was positively correlated with the anti-inflammatory factors *IL-10* and *TGF-β* and inversely correlated with the pro-inflammatory factors *IL-6* and *IL-1β* (Appendix A). In the correlation analysis of intestinal bacteria and serum metabolites (Appendix A), L-tryptophan, LysoPC (20:4 (8Z, 11Z, 14Z, 17Z)), and threoninyl–methionine were significantly positively correlated with *Lactobacillus*, however. LysoPC (20:4 (8Z, 11Z, 14Z, 17Z)) and threoninyl–methionine were significantly negatively correlated with *Akkermansia*.

## 4. Discussion

Gut microbiota is intricately linked to human health and plays an essential function in maintaining physiological equilibrium [33]. Studies have shown that probiotics can alleviate intestinal inflammation and diarrhea by modulating the variety and structure of intestinal flora, affecting the secretion of cytokines, and changing metabolites and metabolic pathways [34,35,36]. Therefore, this study evaluated the impact of *L. plantarum* H-6 and *W. viridescens* J-1 on mice induced with AAD through lincomycin hydrochloride. In the AAD group, the mice showed clinical symptoms such as diarrhea, intestinal flora disorder, elevated levels of pro-inflammatory cytokines, and changes in metabolites and metabolic pathways, indicating that the diarrhea model was successfully established. In comparison to the AAD group, the imbalance in the gut microbiota in *L. plantarum* H-6-treated mice was reversed, the diversity and richness increased, the pathological features of the colon tissue were obviously relieved, the levels of pro-inflammatory factors in the colon tissue were significantly reduced, and the levels of some amino acids in serum were significantly increased. After *W. viridescens* J-1 treatment, the indicators of the mice improved, but the therapeutic effect of *W. viridescens* J-1 on AAD in mice was not as significant as that of *L. plantarum* H-6 treatment.

Due to an imbalance in cytokines, AAD was accompanied by systemic inflammation. Interleukins and tumor necrosis factors play important roles in the immune system [37,38]. Intake of antibiotics could cause abnormal expression of cytokines [39]. In this study, compared with the NC group, a substantial number of inflammatory infiltrations was seen in the colons of mice in the AAD group, whereas *IL-1β* and *IL-6* levels rose dramatically and *IL-10* and *TGF-β* levels decreased significantly. This further demonstrates that the AAD mouse model was successfully constructed. *IL-1β* and *IL-6* are pro-inflammatory cytokines produced by macrophages, and a high production of these cytokines could lead to disease [40]. *IL-6* is mainly secreted by B cells and T cells and, in most cases, promotes inflammation. Many probiotics alleviate intestinal diseases by reducing *IL-6* levels [41]. *IL-1β* promotes intestinal inflammation by destroying the intestinal epithelial tight junction barrier and increasing intestinal permeability [42]. This study shows that the release of anti-inflammatory cytokines *IL-10* and *TGF-β* increased in AAD mice given *L. plantarum* H-6, whereas *IL-1* and *IL-6* levels decreased dramatically. This indicates that *L. plantarum* H-6 can effectively inhibit inflammation and has good probiotic potential, which can be further developed in future research. Tiantian Hu et al. [43] and Xuebing Han et al. [44] reached similar conclusions in their research.

There is evidence that the result of inflammation could not only lead to changes in intestinal flora but also regulate serum metabolites [29]. The serum metabolomics of different treated mice were analyzed, and significant differences were found. The results show that L-tryptophan, LysoPC (20:4 (8Z, 11Z, 14Z, 17Z)), reduced riboflavin, threoninyl–methionine, and N-palmitoyl serine were significantly decreased in AAD mice. Tryptophan has been shown to exert anti-inflammatory effects, and its regulatory pathway is an important regulator of inflammation [45]. In addition, changes in tryptophan and tryptophan metabolic pathways may be associated with probiotic supplementation [46,47]. *Bifidobacterium* could alleviate the proinflammatory immune response by increasing the level of tryptophan [48]. *Lactobacillus reuteri* prevents colonic inflammation by enhancing the production of indole-carboxaldehyde in the tryptophan metabolic pathway [49]. However, only a few probiotics have been shown to regulate intestinal immune function by affecting tryptophan and its metabolic pathways. Additionally, the specific regulatory mechanisms and genes involved are unclear. Therefore, more studies are needed to explore the regulatory effects of different types of probiotics on tryptophan. 

Healthy gut microbiota is a prerequisite for the host to prevent pathogenic bacteria from colonizing the intestine and to perform normal physiological activities. Dysbiosis of gut microbiota presents a significant risk to the host’s health [50]. The results of microbial sequencing showed that the α diversity index of mice in the AAD + LP group was dramatically increased, indicating that the diversity and abundance of intestinal flora were significantly increased as well, which helped improve the stability of the intestinal flora [51]. Therefore, the positive effect of *L. plantarum* H-6 on the treatment of AAD may be achieved through the regulation of gut microbiota balance.

The composition of gut flora is related to the prevalence of illness (such as IBD, AAD, and cancer) [52,53]. Therefore, a detailed taxonomic analysis of the phylum and genus levels of microbial community composition was conducted. At the phylum level, Firmicutes in the AAD group decreased, whereas Bacteroides and Protebacteria increased compared with the NC group. The AAD + LP group reversed this trend, similar to the results of Darong Huang et al. [54]. Protebacteria were the major phylum of Gram-negative pathogens, and these pathogens had high antibiotic resistance, including *Helicobacter* and *Salmonella* [55]. Another study showed that Proteobacteria and Bacteroides were significantly increased in the intestines of mice with lincomycin hydrochloride-induced AAD [23]. In addition, the proliferation of Proteobacteria can lead to inflammatory bowel disease [56]. At the genus level, *Lactobacillus* and *Lachnospiraceae_NK4A136_group* were significantly increased, and *Bacteroides* decreased dramatically in the AAD + LP group compared with the AAD group. More and more evidence has shown that *Lactobacillus* could improve the intestinal environment and promote intestinal health. In the case of TNF-induced intestinal mucosal injury, *Lactobacillus reuteri* alleviated intestinal inflammation by modulating the Wnt/β-catenin signaling pathway to protect the intestinal mucosal barrier and to lower the release of pro-inflammatory molecules [57]. Compared with *Lactobacillus reuteri*, *Lactobacillus planturum* had a stronger inhibitory effect on inflammation [58]. *Lactobacillus planturum* attenuates DSS-induced colitis by improving intestinal inflammation and restoring disturbed gut microbiota [59]. According to Wuyundalai Bao et al., *Lactobacillus plantarum* could balance substance levels and energy metabolism to alleviate AAD [36]. In summary, *Lactobacillus planturum* can treat AAD and intestinal inflammation by regulating the microbial community structure.

Previous studies have shown that intestinal flora regulate peripheral cytokines, which also affect the composition and structure of intestinal flora [60]. In this study, there was a good positive correlation between *Lactobacillus* and *IL-10* and *TGF-β*. It can be inferred that the secretion of cytokines is related to intestinal flora. *L. plantarum* H-6 may alleviate intestinal inflammation in mice by regulating colon *IL-10* and *TGF-β*. Spearman correlation analysis revealed a favorable association between *Lactobacillus* and L-tryptophan content. At present, there is no evidence of whether *Lactobacillus* can affect the concentration of L-tryptophan or whether L-tryptophan can affect *Lactobacillus* count during the development of AAD. To provide a more thorough foundation for the therapy of AAD, further study is necessary to clarify the relationship between L-tryptophan and *Lactobacillus*.

Lactic acid bacteria, as the inherent flora of the host intestinal tract, have good colonization ability in the intestinal tract. Many studies have shown that lactic acid bacteria can regulate the intestinal microenvironment by antagonizing pathogens and releasing nutrients [61], which is consistent with our experimental results. AAD has been plaguing humans and poses a huge threat to human intestinal health. People have been looking for more effective ways to prevent and treat AAD. Studies have shown that a combination of probiotics composed of *Lactobacillus* and *Bifidobacteria* can improve AAD, but the study does not explain why probiotics can alleviate AAD [62]. Yang Chen et al. showed that a novel probiotic beverage (containing *Lactobacillus rhamnosus*, *Lactobacillus acidophilus*, and *Bifidobacterium lactis*) could alleviate AAD by restoring the structure of gut and fecal microbiota [63]. In addition, studies have shown that traditional Chinese medicine (Shen Ling Bai Zhu San) and Poria cocos polysaccharides can also alleviate AAD by regulating intestinal flora [64,65]. It can be seen that previous treatments for AAD have focused on regulating the intestinal flora of the host. In this study, the effects of *L. plantarum* H-6 and *W. viridescens* J-1 on AAD were described from three aspects: intestinal flora, inflammatory factors, and serum metabolites. On this basis, a correlation analysis of the above three aspects was carried out, which helped to elucidate the pathogenesis of AAD. However, in order to be applied to the treatment of clinical patients, we need to explore its mechanism of action further and need to determine the optimal dose of the strain, the frequency of use, and so on.

This study elucidated that *L. plantarum* H-6 treats AAD by regulating intestinal flora and cytokines, as well as changing serum metabolites. This highlights the potential life science value of *L. plantarum* H-6. However, this study has limitations. A mouse AAD model was used to evaluate the probiotic effects of *L. plantarum* H-6, so whether these conclusions can be found in humans remains to be explored. In addition, although *L. plantarum* H-6 was found to have a positive effect on AAD by regulating intestinal flora and cytokines and altering the host metabolism, the regulatory mechanism was not elucidated. The mechanism of *L. plantarum* H-6 alleviating AAD will be a very meaningful topic. However, the effect of *W. viridescens* J-1 on alleviating AAD was significantly lower than that of *L. plantarum* H-6. AAD is caused by gut microbiota imbalance and disorders after antibiotic treatment. According to the results of the study, the ACE index and Chao 1 index of the AAD + WV group did not significantly increase, indicating that *W. viridescens* J-1 treatment did not significantly increase the relative abundance of colon microorganisms in AAD mice. Secondly, AAD is usually accompanied by inflammation, and *W. viridescens* J-1 was less able to inhibit pro-inflammatory factors (*IL-1β*, *IL-6*) and increase anti-inflammatory factors (*IL-10*, *TGF-β*) than *L. plantarum* H-6.

## 5. Conclusions

This study explored the alleviation effect of *L. plantarum* H-6 and *W. viridescens* J-1 on lincomycin hydrochloride-induced AAD. The results show that *L. plantarum* H-6 inhibited ADD-related diarrhea well. At the same time, *L. plantarum* H-6 increased the richness and diversity of colon flora in mice, reduced the expression of *IL-1β* and *IL-6* in colon tissue, and reduced the content of L-tryptophan, LysoPC (20:4 (8Z, 11Z, 14Z, 17Z)), reduced riboflavin, threoniny–methionine, and N-palmitoyl, However, *W. viridescens* J-1 had little effect on AAD mitigation.

## Figures and Tables

**Figure 1 nutrients-15-04603-f001:**
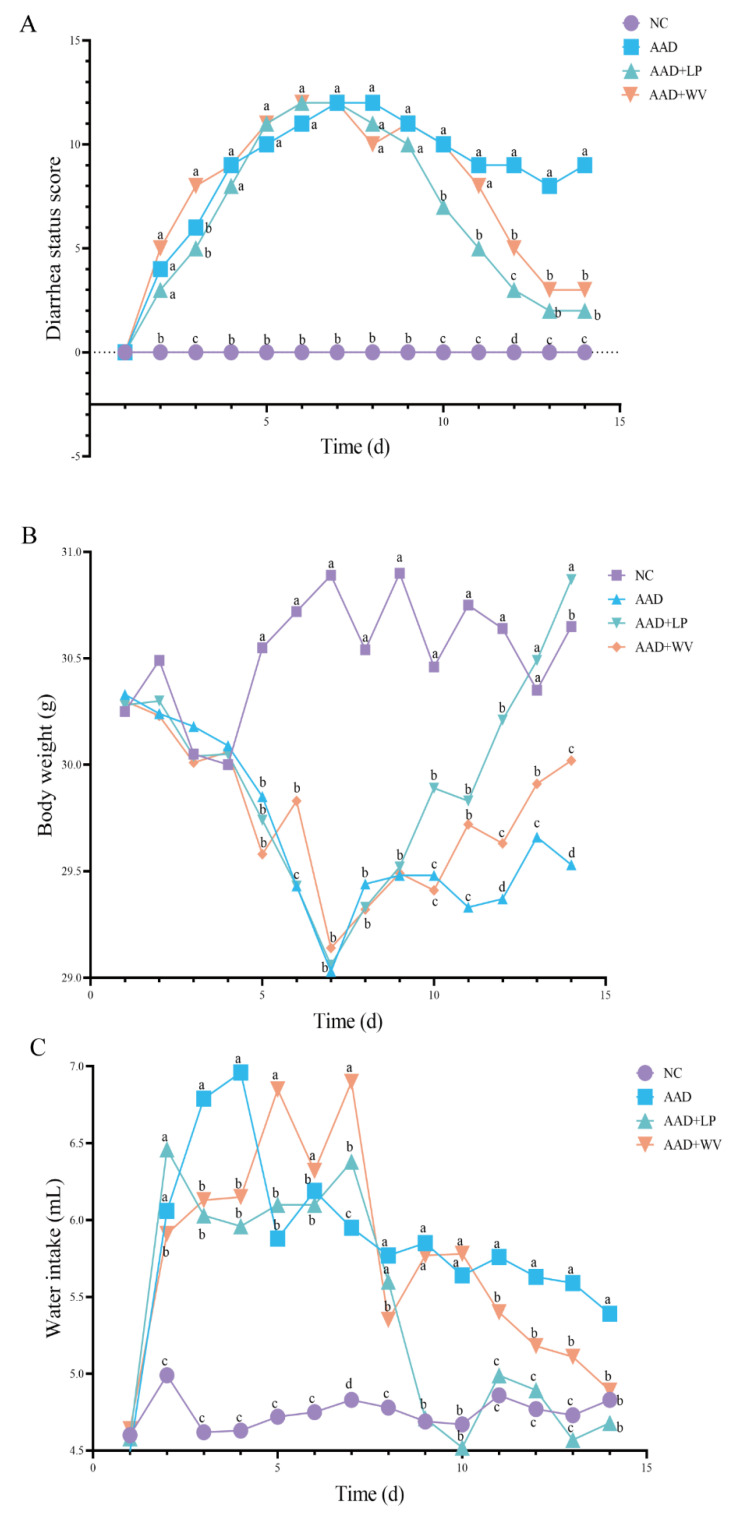
(**A**) Effects of *L. plantarum* H-6 and *W. viridescens* J-1 on diarrhea status scores; (**B**) average body weight; (**C**) average water intake. All data are expressed as average values; different superscripts indicate significant differences at the *p* < 0.05 level on the same day (n = 6).

**Figure 2 nutrients-15-04603-f002:**
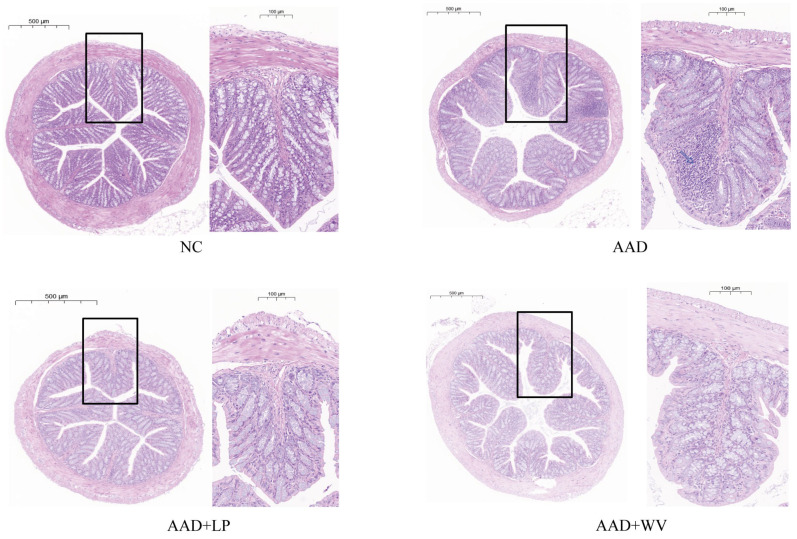
Representative histological images of colon tissues by H&E staining.

**Figure 3 nutrients-15-04603-f003:**
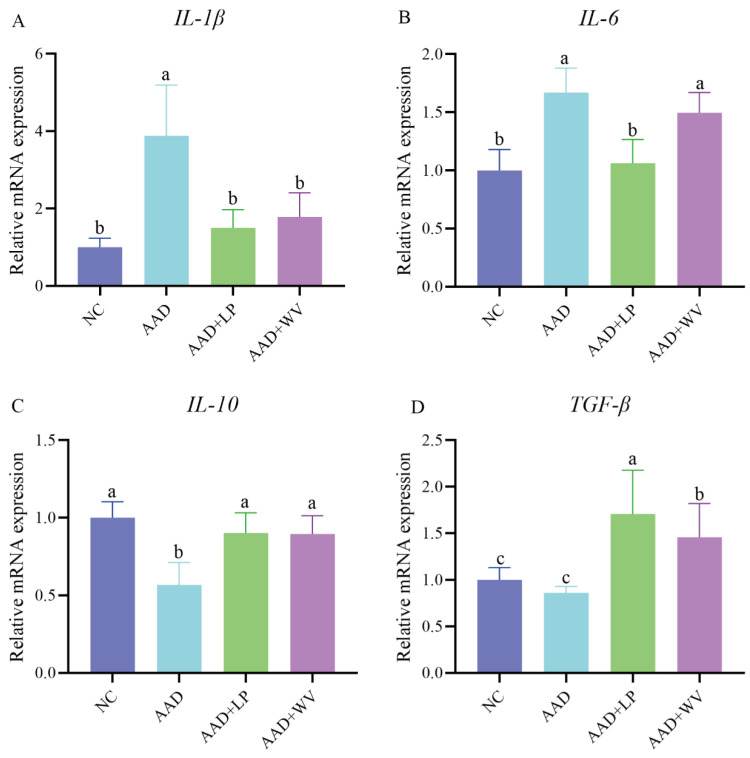
The mRNA expression of inflammatory cytokines in the colon (n = 6), including (**A**) *IL-1β*, (**B**) *IL-6*, (**C**) *IL-10*, and (**D**) *TGF-β*. All data are expressed as mean ± standard errors of the means; values with different superscript letters indicate significant differences at *p* < 0.05.

**Figure 4 nutrients-15-04603-f004:**
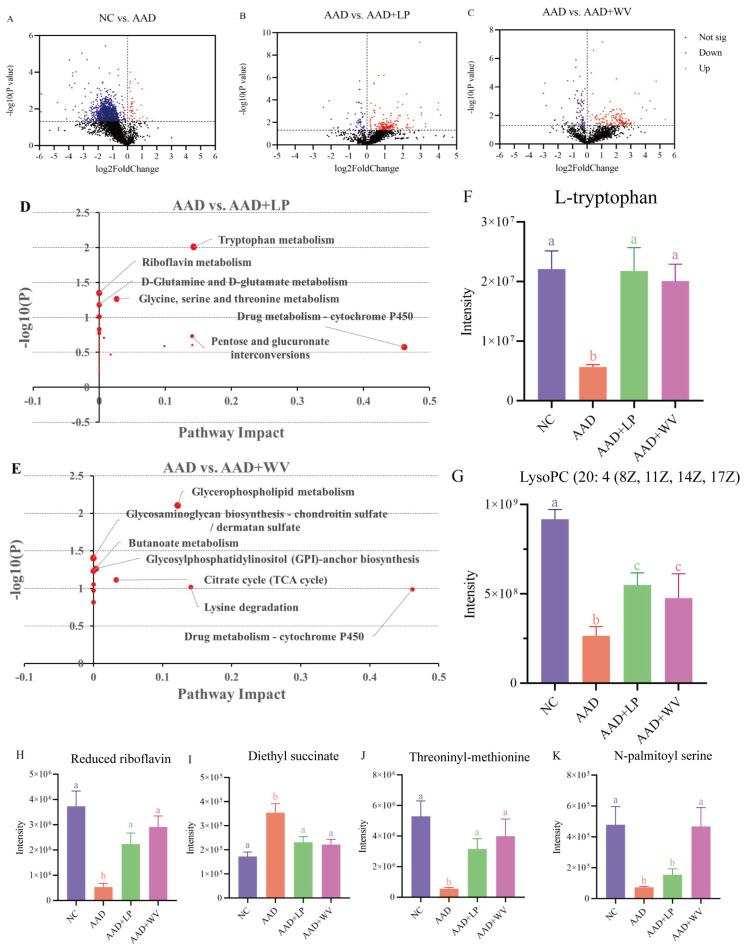
Metabolomics analysis of probiotics-treated LH-induced AAD mice (n = 6). (**A**–**C**) Volcano map of differential metabolites: (**A**) NC vs. AAD; (**B**) AAD vs. AAD + LP; (**C**) AAD vs. AAD + WV. (**D**) The main differential metabolic pathways in *L. plantarum* H-6-treated AAD mice. (**E**) The main differential metabolic pathways in *W. viridescens J-1*-treated AAD mice. (**F**–**K**) The metabolites of serum following *L. plantarum* H-6 and *W. viridescens* J-1 showed significant changes: (**F**) L-tryptophan; (**G**) LysoPC (20:4 (8Z, 11Z, 14Z, 17Z); (**H**) reduced riboflavin; (**I**) diethyl succinate; (**J**) threoninyl–methionine; (**K**) N-palmitoyl serine. All data are expressed as mean ± standard errors of the means; values with different superscript letters indicate significant differences at *p* < 0.05.

**Figure 5 nutrients-15-04603-f005:**
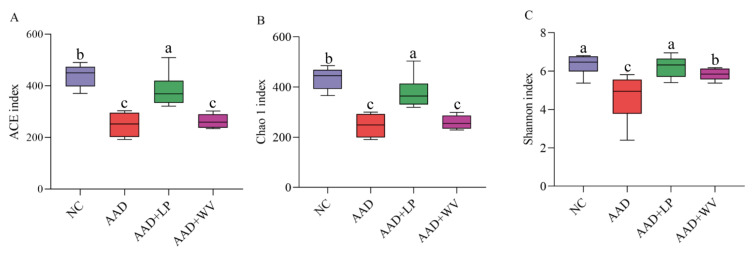
The effects of supplementation of *L. plantarum* H-6 and *W. viridescens* J-1 on the diversity of intestinal flora (n = 6): (**A**) ACE index; (**B**) Chao 1 index; (**C**) Shannon index. All data are expressed as mean ± standard errors of the means; values with different superscript letters indicate significant differences at *p* < 0.05.

**Figure 6 nutrients-15-04603-f006:**
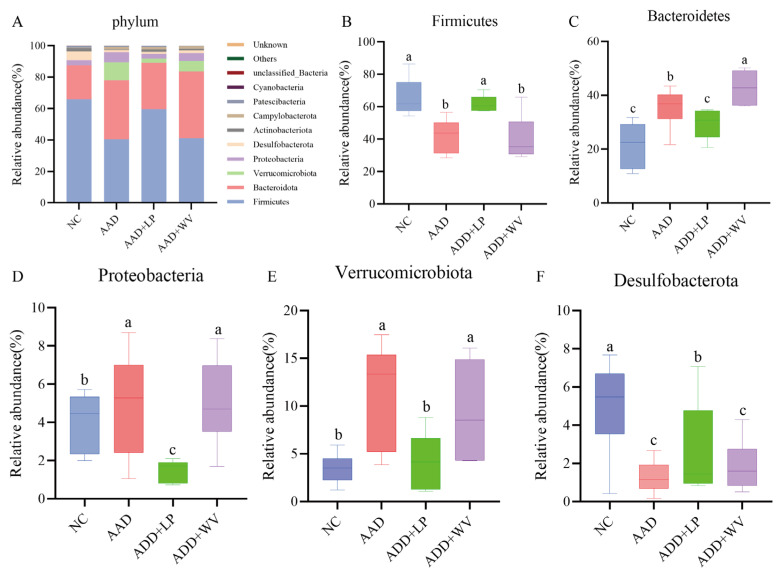
(**A**) Taxonomic compositions of bacterial communities in colonic contents at the phylum level (n = 6). (**B**–**F**) Relative abundance of bacteria in four groups in a sample from each of the four groups: (**B**) Fimicutes; (**C**) Bacteroidetes; (**D**) Proteobacteria; (**E**) Verrucomicrobiota; (**F**) Desulfobacterota. All data are expressed as mean ± standard errors of the means; values with different superscript letters indicate significant differences at *p* < 0.05.

**Figure 7 nutrients-15-04603-f007:**
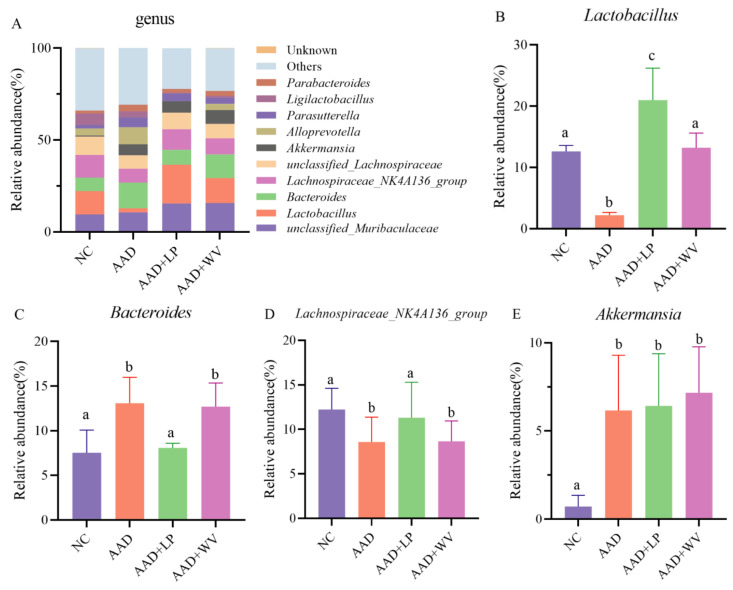
(**A**) Taxonomic compositions of bacterial communities in colonic contents at the genus level (n = 6). (**B**–**F**) Relative abundance of bacteria in four groups in a sample from each of the four groups: (**B**) *Lactobacillus*; (**C**) *Bacteroides*; (**D**) *Lachnospiraceae NK4A136 group*; (**E**) *Akkermansia*. All data are expressed as mean ± standard errors of the means; values with different superscript letters indicate significant differences at *p* < 0.05.

**Figure 8 nutrients-15-04603-f008:**
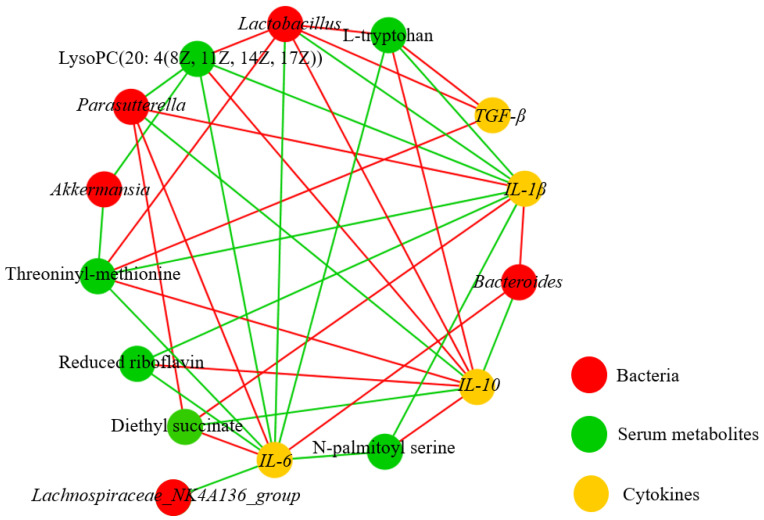
Correlation analysis between metabolites–microorganisms–cytokines (n = 6). The red and green lines represent positive and negative correlation, respectively. The red circle represents genus-level bacteria, the green circle represents serum metabolites, and the yellow circle represents cytokines.

**Table 1 nutrients-15-04603-t001:** Diarrhea status scoring methods.

Scores	Diarrhea Status
0	Mental state was normal, no diarrhea
1	Loose and non-stick perianal stools, average mental state
2	Severe diarrhea, loss of appetite, weight loss, mental malaise

## Data Availability

The raw sequence data in this study are uploaded in the NCBI database; the accession is PRJNA914776 (https://www.ncbi.nlm.nih.gov/bioproject/PRJNA914776) (accessed on 29 December 2022).

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
