# Peer review of "Effects of Lactobacillus plantarum and Weissella viridescens on the Gut Microbiota and Serum Metabolites of Mice with Antibiotic-Associated Diarrhea"

_nutrients, 2023, doi:10.3390/nu15214603_

Round 1
Reviewer 1 Report
This paper is clearly written and interesting to review. Please make changes in text to be uniform (naming amino acids)...when writting strains of bacteria used, don`t italicize strains (H-6 and J-1).
The paper: "Effects of Lactobacillus plantarum and Weissella viridescens on the gut microbiota and serum metabolites of mice with antibiotic-associated diarrhea" by Yan et al is about impact of probiotic bacteria on gut microbiota and serum metabolites in mice with diarrhea (associated with antibiotics). The main question in this research is: are mentioned probiotics, used after antibiotic treatment able to reduce pro-inflamatory factors, decrease harmful gut microflora and improve metabolic processes.
The topic of this research is important and relevant to the field. By contemporary methods, this research addresses specific gap in the field of adverse activity of antibiotics and beneficial effect of probiotics.
The main addition to the subject area compared with other published material are molecular methods and good metabolomics.
The main problem with this research is small number of animals used for the experiment. 24 animals in 4 groups (6 animals in each group) is a problem...the authors should consider either repetition of the experiment or increase number of animals in each group (12 animals per group would be minimum, if no repetition of the experiment).
Conclusions are clearly presented and follow the experimental results.
References used in the paper are appropriate.
Figures are clearly presented and easy to understand.

Reviewer 2 Report
Yan et al. investigated the effects of L. plantarum H-6 and W. viridescens J-1 on antibiotic-associated diarrhea. The authors administered lincomycin hydrocholoride to mice and then treated them with two bacterial strains of interest. They assessed diarrhea by measuring body weight, food consumption, and water intake. Fecal microbial composition, serum metabolic changes, histological changes, and transcriptional changes of cytokines in colon segments were measured. The authors identified L. plantarum was more effective than the W. viridescens in alleviating the AAD.
The following are the comments to this manuscript.
The introduction section was not adequate to explain the purpose of this study. It failed to explain the gaps in the literature that this study aimed to address. When there were clinical trials conducted in human to treat the AAD using probiotic strains, it is important to explain the basis and anticipated outcomes of this study.
The discussion failed to explain the importance of this study’s outcomes and did not adequately compare the outcomes with existing literature. Explaining the benefits of this study to treat AAD in humans is important.
The authors mentioned LEfSe tool was used for microbiome analysis and not clear which of the results was using it.
The low magnification histological images are not sufficient to show the histological changes mentioned in the results section.
The authors should provide the RCF values instead of RPM for anyone to reproduce this study.
The thickness of the tissue slices was given in mol/L.
In several places, appropriate terms were not used to explain the contexts. The overall quality of this manuscript needs to be improved.
